# A Geographical Origin Classification of Durian (cv. Monthong) Using Near-Infrared Diffuse Reflectance Spectroscopy

**DOI:** 10.3390/foods12203844

**Published:** 2023-10-20

**Authors:** Kingdow Chanachot, Wanphut Saechua, Jetsada Posom, Panmanas Sirisomboon

**Affiliations:** 1Department of Agricultural Engineering, School of Engineering, King Mongkut’s Institute of Technology Ladkrabang, Bangkok 10520, Thailand; 61601171@kmitl.ac.th (K.C.); panmanas.si@kmitl.ac.th (P.S.); 2Department of Agricultural Engineering, Faculty of Engineering, Khon Kaen University, Khon Kaen 40002, Thailand

**Keywords:** geographical identification (GI), regions classification, near-infrared spectroscopy, synthetic minority over-sampling technique

## Abstract

The objective of this research was to classify the geographical origin of durians (cv. Monthong) based on geographical identification (GI) and regions (R) using near infrared (NIR). The samples were scanned with an FT-NIR spectrometer (12,500 to 4000 cm^−1^). The NIR absorbance differences among samples that were collected from different parts of the fruit, including intact peel with thorns (I-form), cut-thorn peel (C-form), stem (S-form), and the applied synthetic minority over-sampling technique (SMOTE), were also investigated. Models were developed across several classification algorithms by the classification learner app in MATLAB. The models were optimized using a featured wavenumber selected by a genetic algorithm (GA). An effective model based on GI was developed using SMOTE-I-spectra with a neural network; accuracy was provided as 95.60% and 95.00% in cross-validation and training sets. The test model was provided with a testing set value of %accuracy, and 94.70% by the testing set was obtained. Likewise, the model based on the regions was developed from SMOTE-ICS-form spectra, with the ensemble classifier showing the best result. The best result, 88.00FF% accuracy by cross validation, 86.50% by training set, and 64.90% by testing set, indicates the classification model of East (E-region), Northeast (NE-region), and South (S-region) regions could be applied for rough screening. In summary, NIR spectroscopy could be used as a rapid and nondestructive method for the accurate GI classification of durians.

## 1. Introduction

Durian (*Durio zibethinus* Merr.) is a famous and popular native fruit of Southeast Asia. It is known as the king of fruit due to its rich, sweet, creamy texture, extraordinarily delicious taste, and strong, distinctive odor [1]. Thailand has favorable weather conditions and productive land for the cultivation of tropical and subtropical fruits, such as durian, and the produced durian is tasty, flavorful, and of good quality. Durian is mainly grown in the eastern and southern regions of Thailand. According to an Office of Agricultural Economics report, 24.79% of the durian yield was produced by Chumphon Province, 5.18% by Surat Thani Province, 0.68% by Prachuap Khiri Khan Province (southern Thailand), 24.66% by Chanthaburi Province, 8.39% by Rayong Province, 3.5% by Trat Province, 0.24% by Prajinburi Province (eastern Thailand), and 0.41% by Sri Sa ket Province (northeastern Thailand) [2]. In addition, according to the agricultural product production index, in May 2021, the yield of important agricultural commodities, such as durian and mangosteen, increased compared to April 2021 by 2.55% [2]. In the past, the durian harvesting period in the eastern region was from May to August, and that in the south was from July to August. To satisfy customer demand, the harvesting season has been managed by adding fertilizer and improving cultivation and harvesting methods. As a result, the harvesting periods of provinces or regions might overlap. Therefore, the geographical identification (GI) of durians in Thailand, which is assigned by the Department of Intellectual Property (DIP) under the supervision of the Ministry of Commerce, differs depending on the original province. The GI is a name and symbol representing the geographic origin of a product. It reveals the quality, reputation, or characteristics of the geographical origin of the product, and its determination may be affected by natural and human factors. In addition, GI can represent a local brand to indicate the quality and source of the product. The benefits of registering the GI are product protection, added value, maintaining and preserving local wisdom, sustaining community tourism, and supporting the confidence of buyers in production sources and quality. The GI registration application consists of details of the product regarding quality, reputation, or properties as well as other details, as specified by the ministerial regulations (Geographical Indications Protection Act, B.E. 2546, Article 5–10). In this study, the classification of the geographical origin of durians from Prachuap Kiri Khan Province in southern Thailand was investigated to obtain the GI label. Likewise, for durian plants in Penang, Malaysia, the study indicated that both cultural and marketing terroir differentiate durians from Penang. Two main factors are the way the farmer takes care of trees and the techniques that durian sellers use to select and present the durian to customers [3].

The geographical origin might be used as an indicator to set the price for Thai durians. Because of unique characteristics, durians from some origins must be reserved for harvesting in advance, especially GI durians, and the GI region price must be higher than those in other provinces or regions. Statistically, the average retail price of durians from Prachuap Kiri Khan Province was 300–350 THB/kg, while in other provinces, the price was 200–250 THB/kg. Customers were willing to pay more for high-quality durians if the quality was satisfactory. There is no one who can verify whether the GI label attached to a product is genuine. However, there have been no clear reports of copyright infringement. Therefore, the geographic origin should be reliably identified; otherwise, the false identification of a fruit’s geographical origin could negatively impact the image of farmers. Currently, the discrimination of durian geographical origin is limited to the detection of a few active components with analytical methods such as stable isotopes and element compositions, which is complicated. Additionally, human perception skills are needed. However, human skill is individual expertise. It takes time to transfer this practical knowledge to others, especially to most people who are involved in agriculture or the durian industry. As a result, modern technologies have been introduced and widely used as enabling tools to replace human tasks [1]. However, human skill in distinguishing the geographic origin of durian is not easy to develop by only one way of skill transfer from expertise to unskilled persons, which not only takes long time but also the perceptual ability of the person and a tremendous number of intact fruits and pulps.

Durian has been studied on stable isotope and element compositions, which discriminate the geographic origin of durians from four Southeast Asian countries, namely, Malaysia, Thailand, Cambodia, and Vietnam [4]. Isotope ratios are affected by biogeochemical processes related to the meteorological cycle of evaporation, condensation, and precipitation, which give rise to the spatial variability in hydrogen and oxygen stable isotopes when water is incorporated into plant tissues through photosynthesis and animal tissues through ingestion. This spatial variability is used for food origin and criminal forensic studies. Zhou et al. [4] identified significant differences in tissue type among durians of different geographical origins. The study found that analysis of the durian core provided better origin traceability results than analysis of the durian pulp [4], where durian origin classification (5 regions) achieved percent accuracy rates of 98.60% and 97.30% for the testing and training sets, respectively. An artificial neural network (ANN) yielded higher accuracy rates for identifying durian origin than LDA, namely, 100.00% on the training set and 94.40% on the testing set [4]. Climate or the environment of the geographical origin might cause different metabolite profiles of durian pulp, such as organic acids and amino acids, which are related to flavors [5]. Amino acids also affect fruit quality attributes by acting as precursors of volatile compounds in fruit aromas [6]. Various factors were found to contribute to the quality based on physiochemical and sensory characteristics in geographical regions. These were very important for the consumer’s perceptions of fruit quality and popularity [7], which affected producers [8] in terms of popularity and market prices [9]. The physicochemical characteristics of fruits depend on numerous factors, such as geographic, climatic, and agronomic criteria, which are also used to define product authenticity [10]. To enable producers, consumers, and traders to make unbiased decisions for trading or consuming, it is important to develop a reliable, efficient, and accurate method for discriminating the geographical origins of durians, especially with a nondestructive method.

Visible/infrared spectroscopy and hyperspectral imaging techniques, which are rapid and nondestructive analytical methods, have been widely utilized to trace food varieties and geographical origins [11], such as green coffee beans [12], wines [13], loquats [14], apples [15], and goji berries [16]. The combination of NIR spectroscopy with chemometrics provided an approach for investigating the geographical origins of *Codonopsis pilosula* (*C. pilosula*) in China [16]. This technique of both classification modes performed by RF and K-NN using the PCA scores of samples yielded results with a total accuracy of 94.00% [16]. The NIR spectral range of 1000–2500 nm, combined with LS-SVM, radial basis function ANN (RBF-ANN), PLS-DA, and K-NN, was used to model the classification of *Rhizoma Corydalis* samples, and the use of wavelet transform (WT) input variables from LS-SVM provided slightly better discrimination, with discrimination rates of 100.00% on the calibration set and 95.00% on both the verification set (validation set) and the test dataset [17].

The different locations along level of latitude and longitude in our case—East: Prajinburi, Rayong, Chanthaburi, and Trat; North east: Si sa ket; and South: Surat thani, Chumphon, and Prachuap khiri khan—have a direct effect on geographic characteristics. Durian that comes from different provinces has different tastes, textures, and flavors, which influences consumers and price. The harvesting period of provinces or regions might overlap. This is why these regions were chosen for this research. These variations in agroecological conditions, annual weather patterns, and cultural practices can affect the result, which is why the main aim of this study is to to use the NIR technique to classify different groups of samples, whether they are from different geographical identifications or regions. This study aimed to (1) find an effective model for classifying the geographical identification (GI) and region (R) of durians (cv. Monthong) by (a) applying different spectral pre-processing on the full wavenumber and (b) applying the Synthetic Minority Over-sampling Technique (SMOTE), which is the most common method to address imbalanced classification, which is an important task in supervised learning; (2) select wavenumber ranges by using a genetic algorithm (GA) and develop models by several algorithms (24 algorithms). By MATLAB classification learner and (3) compare the models developed from NIR spectra of intact peels (with thorns) (I-form), cut-thorn peels (C-form), stems (S-form), a combination of I and C (IC-form), a combination of I and S (IS-form), a combination of C and S (CS-form), and a combination of I, C, and S (ICS-form) in terms of % accuracy and sensitivity.

## 2. Materials and Methods

### 2.1. Durian Sample Collection

To ensure geographical origin, durian samples were directly collected from various orchards in different provinces and regions of Thailand. Durian samples were received from the East: Chanthaburi, Prajinburi, Rayong, and Trat provinces; the South: Chumphon, Surat thani, and Prachuap Kiri Khan (GI durian) provinces; and the Northeast: Sri sa ket province. The details of the collection of 120 durian fruits are presented in Table 1 and Figure 1. For the sampling process, managing the variation of sampling was done by the following: (1) collected 3 orchards per province. (2) Durian tree age of 5–10 years, which is covered in the production year (3) 2 years for collecting samples (2019–2020), indicated the replication of sampling to include variation effects of agroecological conditions and annual weather patterns. (4) All durians were tagged on 120 DAA (day after anthesis), which is the commercial harvesting date. The tagging sampling was 3 durian fruits, and we selected 1. (5) The durian sample was tagged in the middle level of the canopy. (6) Transportation was controlled within 3 days, and the fruit samples were controlled for uniform temperature during scanning to avoid spectral error in the experiment. One hundred and twenty durian fruits were first collected and used for model creation and model proving. The collection conditions of the durian samples were determined by the National Bureau of Agricultural Commodity and Food Standards [18] as follows: The harvesting date was 120 days after anthesis (DAA), and the weight of each fruit was between 2.50 and 3.50 kg. Durian trees were irrigated with water, fertilizers, or/and chemical treatments guided by durian fruit experts. The samples were transported directly to the NIR Spectroscopy Research Center of Agricultural Product and Food, Department of Agricultural Engineering, School of Engineering, King Mongkut’s Institute of Technology, Ladkrabang, Thailand. After that, the samples were kept in the air at room temperature (25 ± 2 °C) overnight before the experiment for approximately 20 h.

### 2.2. NIR Spectral Acquisition

NIR spectral data were acquired over a wavenumber range of 12,500–4000 cm^−1^ in diffuse reflectance mode using an FT-NIR spectrometer (Multi-Purpose Analyzer, MPA, Bruker Optics, Bremen, Germany), with a scanning resolution of 16 cm^−1^ and 32 scans for each spectrum. Each durian fruit was scanned repeatedly three times at the same point, using all the data in the data analysis in 3 different forms: I-form, C-form, and S-form. The I-form and C-form were scanned in the middle of each locule of the whole durian fruit, and a black sponge was used to cover the gap between the sample and the NIR beam window of the spectrometer to avoid leakage of NIR radiation. The stem was scanned on 2 sides at a 180° distance in the middle part of the stem. Before scanning every form and sample, a gold plate was scanned internally for background compensation. Spectral data were collected and analyzed using OPUS 6.5 (Bruker Optik GmbH, Germany). The measurement setup, configuration of sample scanning, and scanned position are shown in Figure 2.

### 2.3. Classification Modeling

After spectral collection, each NIR spectrum and its corresponding GI and R categories were registered for model development. A flow chart of the study is shown in Figure 3. There were 2 geographical origin categories (GI and R). The dataset of GI consisted of 2 classes, namely, GI (durian samples received from Prachuap Kiri Khan province) and non-GI, and the R dataset consisted of 3 classes, namely, East (E-region), Northeast (NE-region), and South (S-region) regions.

The accuracies of the developed models with different durian forms, namely, I-form, C-form, S-form, I+C (IC-form), I+S (IS-form), C+S (CS-form), and I+C+S (ICS-form), were compared. The spectral combination strategy is from the study of Somton et al. (2015) The study was used to classify the durian samples into immature, early mature, and mature classes based on the number of days after anthesis. A combination of both rind and stem spectral data provided the highest accuracy of classification at 94.40% [19], and Sirirak et al. (2023) found that the performance of the model increases when developed with a combination of rind and stem spectra [20]. The non-destructive maturity classification model of Monthong durians is based on the visible spectroscopy of durian husk. The data discriminant analysis showed that model can be separated into five groups, which have 83.3% accuracy [21].

All spectra from different durian forms (I-form, C-form, S-form, IC-form, IS-form, CS-form, and ICS-form) and either raw spectra or pre-processed spectra (standard normal variate (SNV), baseline offset (Baseline), multiplicative scattering correction (MSC), detrending (Detrend), mean normalization (MeanNor), Savitzky-Goley first derivative (FD), and Savitzky-Goley second derivative (SD)) were used for model development. It was necessary to perform mathematical pre-processing to reduce systematic noise, such as baseline variation and light scattering, and to enhance the contribution of the chemical composition [16]. The concentrations of the chemical constituents and the physical properties, such as density, particle size, and particle distribution, which can change the path length and absorbance of the sample, affect the NIR spectra. Savitzky-Goley smoothing is an effective approach for removing high-frequency noise from a spectrum and improving the signal-to-noise ratio, while detrending is an approach for eliminating the baseline shift in the spectrum [22]. In the case of transmission measurement, normalization is used to eliminate the influence of different optical path lengths, which can only change the height of the signal but not its structure. Likewise, in the case of the diffuse reflectance model, partial size influences the NIR spectra [23]. MSC can be effectively used to eliminate the translation and offset of the baseline [23]. MSC is used to compensate for the effect of nonuniform scattering induced by diverse particle sizes, uneven distributions, and other physical effects in the spectral data. SNV basically performs the same as MSC [24]. The objective of SNV is to eliminate the deviations caused by particle size and scattering and to eliminate drifting and scattering [22]. FD and SD are used to remove background interference, distinguish superimposed peaks, and enhance the spectral resolution and sensitivity [22]. Additionally, derivative pre-processing leads to more stable reflectance, but it is sensitive to noise; thus, it needs to be used with a smoothing algorithm [24].

The classification learner app of MATLAB was used to develop a classification model using supervised machine learning methods [22]. Various classification models based on statistics and machine learning in the classification learner of MATLAB R2021b (license no. 40846673, MathWorks, Natick, MA, USA) were applied. A total of 24 algorithms were compared. The algorithms were from six main classification groups: A decision tree is a flowchart similar to a tree structure, where each internal node represents a test on an attribute, each branch represents an outcome of the corresponding test, and each leaf node (terminal node) holds a class label. Decision tree types consist of simple, medium, and complex.Naïve Bayes is a collection of classification algorithms based on Bayes’ Theorem, which is related to probability. There are two types of Naïve Bayes: Gaussian Naïve Bayes and kernel Naïve Bayes.A support vector machine (SVM) is a mathematical function used to determine the limits between classes [25] by using a hyperplane and margin lines. The available types of SVMs are linear, quadratic, cubic, fine Gaussian, medium Gaussian, and coarse Gaussian.K-nearest neighbors (K-NN) is used to measure the distance between K-nearest neighbors. K-NN consists of 6 types: fine, medium, coarse, cosine, cubic, and weighted.Discriminant analysis is used to find combinations of features that can characterize or separate classes. Discriminant analysis can be linear or quadratic [25].Ensemble classification is a meta-algorithm that can combine several machine learning techniques into one predictive model to decrease variance or bias or improve predictions. There are several types: bagged trees, boosted trees, subspace discriminant, subspace K-NN, and RUSBoosted trees.

There were two ways of the datasets were split: (1) cross validation and (2) training and testing set. Cross validation: to avoid overfitting, the classification model was validated using repeated nested cross-validation. An advantage of nested cross-validation is the estimation of the true error, which can be almost unbiased [26]. K-fold cross-validation is a common cross-validation technique in which the data are partitioned into K randomly chosen subsets (or folds) of roughly equal size. One subset is used to validate the model trained using the remaining subsets. This process is repeated K times, such that each subset is used exactly once for validation [27]. A fivefold outer cross-validation was performed by repeating the process five times to assess the model performance. Training and testing set: sample selection, the samples were divided into training (75%) and test (25%) sets using the Kennard-Stone algorithm, which is based on a Euclidian distance calculation, where the sample with a maximum distance to all other samples is selected, then the samples that are as far away as possible from the selected samples are selected, until the selected number of samples is reached. This means that the samples are selected in such a way that they will uniformly cover the complete sample space, reducing the need for extrapolation of the remaining samples [28]. 

The second cycle of analysis was comparing SMOTE and original data (the imbalance data set). SMOTE is the most common method to address imbalanced classification, which is an important task in supervised learning [29], including in NIRS [30].

After model creation, the testing set was analyzed. The model with the highest accuracy on the training set and test set was selected as the optimal model. In summary, the pre-processed spectral data of each form selected by optimal models will be used as input spectral data for the wavenumber selection process.

### 2.4. Optimal Wavenumber Screening and Optimization

After model creation and validation by testing samples, the spectral pre-processing that yielded the highest accuracy was selected. It was assigned as an effective spectral pre-processing method, which was used in the GA to determine the most important relationship between geographical origin and wavenumber. 

A practical approach for improving the model’s robustness and accuracy is the elimination of irrelevant variables and redundancies in the data and the selection of relevant spectral features [31]. The selected wavelength method can reduce the number of independent variables to obtain only the most significant spectra consisting of features of interest. Feature selection reduces the spectral complexity and selects only useful wavelengths that are highly correlated with the predicted value. GA is an adaptive technique that can be successfully used in solving complex search and optimization problems [32]. The GA is a variable selection method that can effectively solve the problem of spectral collinearity and has the great advantages of high repeatability and minimal redundancy of spectral information. Theoretically, the GA is a type of mathematical model inspired by Charles Darwin’s idea of natural selection. Simple GA pseudocodes consist of (1) choosing the initial population of individuals, (2) evaluating the fitness of each individual in that population, and (3) iterating on this generation until termination (time limit, sufficient fitness achieved, etc.) [33]. Therefore, wavelength selection is needed before calibration.

Then, classification models were developed using wavenumbers selected by the GA within 24 algorithms again. Finally, the models developed using the full wavenumber range and the wavenumbers selected by the GA were compared in terms of accuracy.

### 2.5. Evaluation of Model Performance

The NIR spectra test set was used to evaluate the accuracy, precision, sensitivity, and F1 score, which are the common criteria used to evaluate the performance of classification models. The accuracy percentage is the total correct classification percentage relative to all cases (correct and incorrect). Sensitivity and specificity are common figures of merit used to evaluate classification and authentication models [34]. The sensitivity of a learning machine is the ratio between the number of true-positive predictions and the number of positive instances in the test set (true positive + false negative) [35]. Specificity indicates the correctness of the model in classifying other-group samples (true-negative samples) among all other-group samples (true negative + false positive) [34]. Precision is the ratio of the number of samples correctly classified as model samples (true positive) to the total number of samples classified as model samples (true positive + false positive) [34]. TP is the number of true-positive samples, which are samples that are correctly classified as model samples; TN is the number of true-negative samples, which are samples that are correctly classified as other-group samples; FP is the number of false-positive samples, which are samples from another group that have been classified as model samples; and FN is the number of false-negative samples, which are model samples that have been classified as other-group samples. Equations (1)–(5) present the evaluation parameter formulas:(1)$$\%Accuracy=\frac{TP+TN}{TP+TN+FP+FN}\times 100$$
(2)$$\%Sensitivity=\frac{TP}{TP+FN}\times 100$$
(3)$$\%Specificity=\frac{TN}{TN+FP}\times 100$$
(4)$$\%Precision=\frac{TP}{TP+FP}\times 100$$
(5)$$F1\;score=\frac{2\times Precision\times Sensitivity}{Precision+Sensitivity}$$

The F1 score, the harmonic precision and sensitivity mean, is a classification accuracy metric that combines precision and sensitivity. It is designed to be a useful metric when classifying between unbalanced classes or other cases where simpler metrics could be misleading. It indicates the meaning of correctness in identifying every sample as the true positive and in identifying the true positive as being correctly classified. F1 score is maximum, which is equal to one if precision is equal to sensitivity. 

To find an effective classification model using the Classification Learner app, the model with the highest accuracy and sensitivity on the cross-validation set (validate), or training set and testing set, was selected as the effective classification model. The performances of the model using the full wavenumber range applying SMOTE, selecting featured wavenumbers using GA applying, pre-processing methods, and development with various modelling algorithms were compared.

## 3. Results and Discussion

### 3.1. Spectral Investigation

The average near-infrared (NIR) spectra of durian fruits in three forms (I-form, C-form, and S-form) are graphically presented in Figure 4. The average raw NIR spectra ranged between 12,500 and 4050 cm^−1^. All spectra were similar in terms of overall pattern, and several observed peaks were also similar among all forms of durian samples. The profile curves of the raw spectra of the I-form and C-form were very similar visually. Along the wavenumbers of the spectra, the S-form absorbed less radiation than the I-form and C-form. The difference in scanned surface characteristics could have caused the variation in scattering [19]. The surface of the cut-thorn peel (C-form) was smoother than that of the I-form. This smooth surface produced less reflectance and, thus, more absorbance. Although the complexity of ingredients in natural products such as agricultural products discourage specific chemical group assignments, obvious bands of average raw spectra at 10,314, 8598, 6827, and 5122 cm^−1^ were assigned to the bond vibrations of H_2_O [36]. Sharma et al. (2022) determined the peak wavelengths of durian pulp [37]. The peak at 5639 cm^−1^ corresponded to a vibration due to the C –H str. first overtone of cellulose [38]. All details are presented in Table 2. Durian rind (peel) with thorns contains lignin and is rich in hemicelluloses [38], moisture, ash, carbon, and oxygen [38]. A study on the chemical composition of pasteurized crushed durian peel found that the composition of durian peel consisted of moisture (85.39 ± 0.17%), protein (0.52 ± 0.00%), lipid (0.01 ± 0.01%), fiber (12.86 ± 0.11%), ash (0.35 ± 0.03%), and carbohydrate (0.87 ± 0.06%).

The study of Somton et al. (2015) indicated that NIR absorption by water in the NIR spectra of durian rinds could predict the level of maturity and directly influence the dry matter content (%DM) and sweetness (%Brix) of pulp [19]. Therefore, the farmers used processes to improve the fertility of the rind and/or stem because it might affect the durian pulp. In general, the main functions of the stem consist of (1) transporting water and minerals from the root to the leaves for photosynthesis and (2) transporting glucose to the leaves to produce plant energy, which is necessary for growth. The durian rinds and stems are influenced by several factors. The addition of plant hormones and chemicals was controlled by the farmer, but rainfall, sunlight, storms, and weather could not be controlled. This relation indicates that the stem, rind, and durian pulp are connected. In addition, the I-form, C-form, and S-form NIR spectra indicated that different chemical compositions of durian caused different NIR absorption values. The indication of the relationship between intact durian fruit peel and thorn spectra with the quality of pulp (dry matter, DM) has been shown in the study [38], and the relationship between intact durian fruit rind spectra combined with fruit stem spectra and the quality of pulp (DM) has been shown in Ditcharoen et al. (2023) [20]. Onsawai et al. (2021) [38] indicated with spectral collection methods that scanning of the intact fruit at the largest locule showed R^2^ between NIR spectra and DM of pulp and RMSEC values of 0.80 and 5.19%, respectively, and the predictive ability of the DM model validated using the validation set provided r^2^, RMSEP, and RPD values of 0.79, 5.23%, and 2.18, respectively. These results confirm that the largest locule is the optimum NIR scanning position of the intact durian fruit to acquire spectra for predictive modeling of the DM content of the pulp. Ditcharoen et al. (2023) proved that the combination of rind spectra and stem spectra could improve the maturity classification accuracy based on different DM of durian pulp. The result indicated that the combined spectra (rind and stem) can increase the model performance of all algorithms. The LDA model developed with the rind (D2) + stem (MA + SNV) provided the best classification efficiency, precision, recall (sensitivity), specificity, and overall accuracy of 96.40%, 91.88%, 97.55%, and 97.28%, respectively. These indicated the pulp DM was related to the chemical constituents in the rind and stem of durian fruit. In addition, postharvest biology of durian from Chumphon and Chanthaburi provinces 120 days after pollination included DM (%), soluble solids content (%), and lipid content (g/100 g fresh wt.) of the aril (durian pulp), which indicated a value difference between two provinces. Sangwanangkul et al. [40] studied a correlation between the chemical compositions of the fruit stem and the maturity of ‘Monthong’ durian fruit, and it was shown that the total non-structural carbohydrate (TNC) of the upper fruit stem was significantly correlated with the dry matter of the pulp. This study aimed to analyze the genetic and relatedness of indigenous durian (*Durio zibethinus* Murr.) in southern Thailand using random amplified polymorphic DNA (RAPD) and microsatellite markers. In this study, 67 samples were collected from Songkhla, Nakhon Sithammarat, Krabi Phanga, and Yala provinces in southern Thailand. Resulting from a dendrogram analysis, four clusters could be separated with a genetic similarity index. Cluster corresponds well with their geographical original. Therefore, it is possible to differentiate the geographical origin of durian due to the taste of durian pulps, which is related to DM. However, it is a destructive test. 

Figure 5a shows the average raw spectral data and pre-processing spectra by SD of GI and non-GI in the I-form. There were five obvious peaks related to the chemical composition of the I-form at wavenumbers of 10,391, 8601, 6912, 5585, and 5076 cm^−1^. Most of the peaks were associated with water [41], and the expected peak at 5585 cm^−1^ corresponded to a combination of vibrations due to C–H str. of the first overtone of cellulose [36]. The raw NIR spectral data showed a baseline shift, which was due to different scattering-influenced optical path lengths. The best pre-processing method will be the one that finally produces the best predictive model. Unfortunately, there seem to be no hard rules for deciding which type of pre-processing to apply, and often the only approach is trial and error [42]. However, SD pre-processing was applied to remove background interference, distinguish superimposed peaks, and enhance the spectral resolution and sensitivity [22]. From Figure 5a, the spectra of the GI and non-GI classes were similar. A small difference in the highest peak was observed at 5222 cm^−1^, which was related to a combination of vibrations due to the O–H str. +2 × C–H str. of starch [41]. The peak at 7089 cm^−1^ corresponds to the O–H str. first overtone of ROH, and the small peak at 4397 cm^−1^ might be a shifting peak of 4405; it corresponds to the O–H/C–H of cellulose [41]. Figure 5b shows the average raw spectral data and pre-processed data by SD of the three regions of the ICS-form. The dominant peaks were the same as those in Figure 5a. The interesting observation is the baseline offset. The absorption peaks of water at 6129 and 5076 cm^−1^ clearly showed different levels. The NE region showed the highest absorption, followed by the E-region, and the S-region showed the lowest absorption. Differences in the absorption peak of water between classes might affect the model classification performance. The indicated variables of each class were used to study their contributions to the classification model. The differences in the NIR spectra were mainly related to the chemical composition differences. 

However, many factors influence the differences in durians from different geographic origins, including the effects of season, climate, soil minerals, irrigation, fertilizer, and planting method. Regarding the use of fertilizer in the planting method, the effect depends on the type, pH, and water content of the soil. Normally, the farmer will adjust the formula of the fertilizer by increasing the amount of potassium (K) before harvesting, which results in a higher starch content [43]. This might affect the chemical composition of the durian pulp and rind. This was confirmed by the study of Youryon et al. (2022), which observed aril softening and the conversion of starch to sugar one or two weeks before fruit maturity (the maturity date of durian is 120 DAA [44]. The season and climate are related to the rainfall, temperature, %humidity, etc., which have significant effects on the taste of durian pulp [45] and might also affect the constituents of the rind or peel. Similarly, a study of starch in cassava in Columbia indicated that increasing rainfall in different areas affected the content of starch. Additionally, light affects the synthesis and accumulation of carbohydrates and nitrogen metabolism in plants, soil affects the absorption of mineral elements in plants, and climate affects the growth cycle of plants, resulting in inconsistencies in fruit maturity [46]. Other factors are sunshine hours, poor sunlight, and low temperatures during fruit growth, resulting in a slow accumulation of starch in the aril. This change might be a process that involves physicochemical changes of the durian rind. Durian fruits from trees were shaded by 50% for one week before uneven ripening and harvesting [1]. A study of durians in postharvest biology and technology of tropical and subtropical fruits indicated that durian fruits harvested during a prolonged period of cloudy skies may have less nutrient accumulation in the aril and have poor ripening characteristics [47]. The inputs of photosynthesis of plants are carbon dioxide, water, energy from the sun, chlorophyll, and leaves, which performed this process in which the outputs were sugar and water, which were used for plant growth in the stems, stalks, twigs, and leaves. The accumulation of nutrients influences the starch content. These inputs for photosynthesis in different provinces or regions influenced the chemical composition of the rind, stem, and/or pulp of durian fruits. They all led to changes in the types and contents of the constituents in durian fruits of different geographic origins.

Many studies on agricultural products have indicated a correlation between peel composition and geographical origin. A study of the effect of geographical location on Mexican lime (*Citrus aurantifolia*) peel components indicated that the relative concentration of compounds was different according to the type of location [48]. A study of polyphenolic compositions and antioxidant activities by Kam et al. (2013) identified useful parameters for differentiating the methanolic extracts of pomegranate peel from China, the USA, and Australia. A correlative study provided a valuable method for differentiating the origins of pomegranate peel from different countries [46]. In addition, “*Namdokmai*” mango (*Mangifera indica*) trees grown in Pakchong, Nakhon Ratchasima Province, and Kamphaengsaen, Nakhon Pathom Province, were studied by Zhu et al. (1994). The study indicated that even the same number of days after anthesis, origin, and harvesting time affected the percentage of starch in mangoes [49]. In the same way, Sangwanangkul et al. (1990) studied the physical and chemical compositions of durians (cv. Monthong) harvested in Chanthaburi and Chumphon Province. This result indicated that durian fruit length in the horizontal direction, diameter, weight of fresh fruit, and soluble solid, β-carotene, and fat contents differed between these two provinces, especially in the last three months before commercial consumption. The amount of chemicals in durian peel or stem may be directly related to the geographical origin.

### 3.2. Comparison of Model Performance

Spectral data for all forms (original or SMOTE), including the I-form, C-form, S-form, IC-form, IS-form, CS-form, and ICS-form, were combined with all pre-processing methods, and models were created using 24 algorithms in the classification learner app of MATLAB. Table 3 shows the accuracy values of the effective model for the geographical origin classification of durians by GI and R. According to the results of the study, based on GI overall, SMOTE could improve accuracy in each form. Accuracy ranged from 76.50% to 95.60%. The highest accuracies were 95.60%, 95.00%, and 94.70% for the cross-validation, training set, and testing set of the SMOTE-I form with the SNV pre-processing method, respectively. Based on R, the accuracy range of cross-validation was 56.10–88.00%. Pre-processing with the SNV applied in the SMOTE-ICS-form provided the highest accuracy at 88.00% for cross-validation and 86.50% and 80.00% for the training set and testing set, respectively. It was found that the classification based on GI showed better prediction than that based on R.

The effective model was compared between the full wavenumber range (original or applied SMOTE) and the wavenumbers selected by the GA with a similar algorithm. The selected wavelength by GA was chosen from 1154 wavenumbers. Overall, the developed models using the full wavenumber range (1154 variables) achieved lower accuracies, indicating that the 100 variables selected by GA covered most of the explained variance and that the unselected variables were not related to the GI or R classification. The confusion matrix and classification performance of the best effective models are presented in Table 4. Precision, sensitivity, specificity, and F1 score were shown. Precision is the ratio of the number of samples correctly classified as model samples. Precision will only be considered for positive sample group. Sensitivity is concerned with correctly identifying positive cases, while specificity focuses on correctly identifying negative cases. The F1 score provides a single value that summarizes the trade-off between precision and sensitivity, and it can help mitigate these risks by striking a balance between identifying items with a certain origin and avoiding false identifications [34]. The most effective model of geographical origin classification of durian in terms of GI used classifier-type narrow neural network algorithm with a SMOTE-I data set with SNV pre-processing. The results of the test sets of GI and non-GI show a high percentage of precision, sensitivity, and specificity and a f1 score that is more than 93%. It indicated reliability from every point of view. The case of R classification used an ensemble classifier-type subspace discriminant algorithm with a SMOTE-ICS data set with SNV pre-processing. The highest percentage of precision, sensitivity, and specificity is observed in the E (E-region) class. Its model is acceptable to continue updating. In contrast, the models in the NE (NE-region) class and S (S-region) class were interchanged prediction i.e., NE was more or less predicted to be S and vice versa. The highest number of negative groups of NE-class caused the lowest specificity. This result, as expected, the spectra, which were not as different as they should be, could cause a low percentage of prediction performance. (accuracy, precision, sensitivity, specificity, and F1 score). The similarity in spectral characteristics across various regions in our case, BE and S, depended upon numerous influencing factors (geology, altitude, rainfall, weather (temperature), distance from the sea, etc.), which will be analyzed in a future study. Neural networks were applied to classify groups, as in the study of Kim et al. (1999), where they were used to determine the geographic origins of potatoes [50]. The samples were separated into two geographic regions (defined as Idaho and non-Idaho) with a high percentage of correct classifications and an accuracy value of nearly 100.00. In a study by Khalafyan et al. (2019) [12], the wine variety and geographical origin of white wines were successfully determined using a neural network algorithm. The accuracy of the classification into wine variety groups ranged from 83.00 to 100.00 on all sets of data (training and test sets). Likewise, the most effective model of geographical origin classification of durian in terms of R used ensemble classifier-type subspace discriminant algorithm with SMOTE-ICS data set with SNV pre-processing. The accuracy was 88.00% and 86.50% in the cross-validation set and the training set, respectively. There were a significant number of false classifications observed across all classes. It affected the classification of geographical regions and resulted in an accuracy of 80.00%. For the testing set, the sensitivity measure was found to be merely 64.19%. The primary contributing factor was misclassification, specifically within the NE class. This result indicated that the effective model could be improved by increasing the number of samples in the specific regions.

The best effective classification modeling parameters defined for each model are listed in Table 5. However, the NIR spectra of samples may not reveal dominant differences due to different origins, but with mathematical manipulation, the latent interference was eliminated, and the specific geographical origin featured variables were appealed. The selected absorption wavenumber was proven to improve the classification accuracy.

## 4. Conclusions

An effective model for classifying GI (Prachuap Khiri Khan) from other locations was developed using SMOTE-I-spectra, SNV pre-processing, and a narrow neural network architecture. The model achieved an accuracy of 95.60% and 95.00% in the cross-validation and training sets, respectively, while achieving an accuracy of 94.70% on the test set. Likewise, a geographical origin classification model to classify different regions (R) was developed from SMOTE-ICS-form spectra with SNV, and an ensemble classifier-type subspace discriminant showed the best result, with 88.00 accuracy by cross-validation, 86.50% by training set, and 64.90 accuracy by testing set obtained, indicating the classification model of East (E-region), Northeast (NE-region), and South (S-region) regions could be applied for rough screening [51]. The sensitivity in each geographic region included 93.24%, 64.19%, and 82.43%, respectively. The low performance of the model predicted using an unknown set was due to too few samples, indicating that more samples in an unknown set should be more balanced in all graphical origin groups. NIR diffuse reflectance spectroscopy coupled with chemometrics showed the possible advantages of fast analytical speed and nondestructive measurement for identifying and classifying durians by geographical identification for GI but not for different geographic regions in Thailand in this study. Therefore, the NIR technique could be updated by increasing the number of samples in specific regions, and a more effective wavenumber selection method should be explored to obtain an effective protocol to guarantee the geographic origin of durians in Thailand, which may be applicable to durian plantations in different regions of the world. The geographical origin classification in this case, GI of Prachuap Khiri Khan, can confirm the customer buying intention of unique characteristics. Therefore, it can be concluded that this technique can verify whether the GI label attached to a product is genuine. In the same way, the geographic origin should be reliably identified; otherwise, the false identification of a fruit’s geographical origin could negatively impact the image of farmers.

## Figures and Tables

**Figure 1 foods-12-03844-f001:**
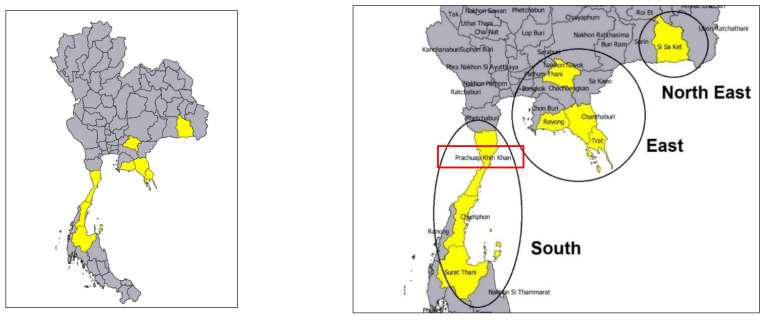
The geographical origin of collected durian sample.

**Figure 2 foods-12-03844-f002:**
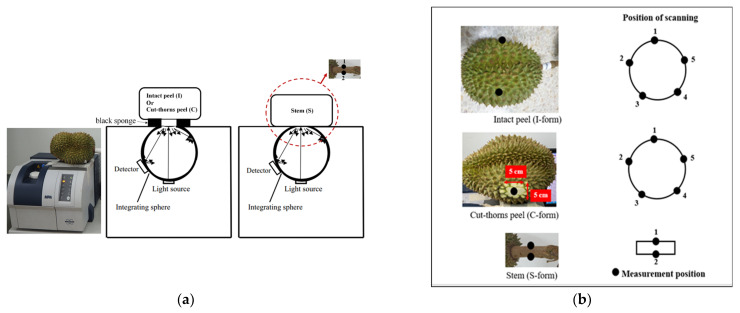
Scanning configuration of FT-NIR spectrometer using 3 forms: intact peel (with thorns) (I-form), cut-thorn peel (C-from), and stem (S-form); (**a**) position views of scanning (**b**).

**Figure 3 foods-12-03844-f003:**
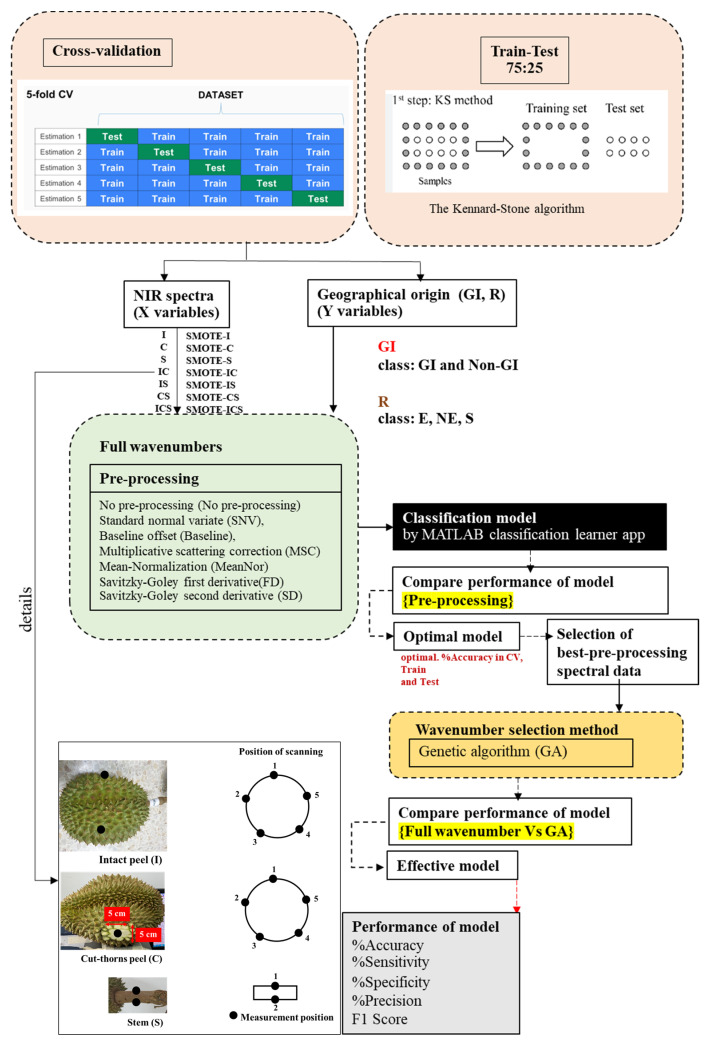
Workflow of the study.

**Figure 4 foods-12-03844-f004:**
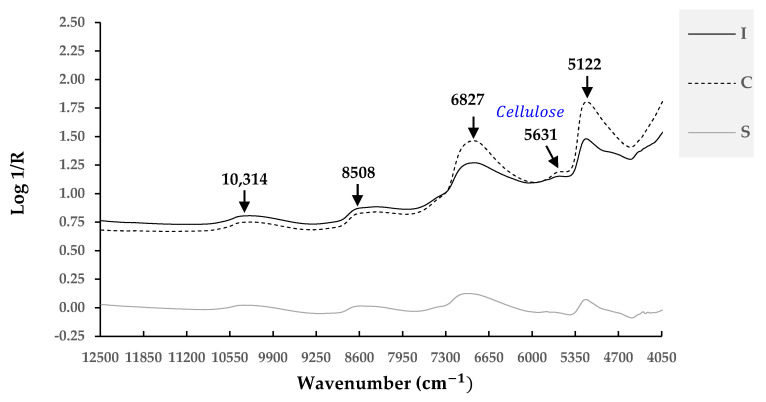
Average raw near-infrared (NIR) spectra of durian fruits in 3 forms: intact peel (with thorns) (I-form), cut-thorns peel (C-form), and stem (S-from).

**Figure 5 foods-12-03844-f005:**
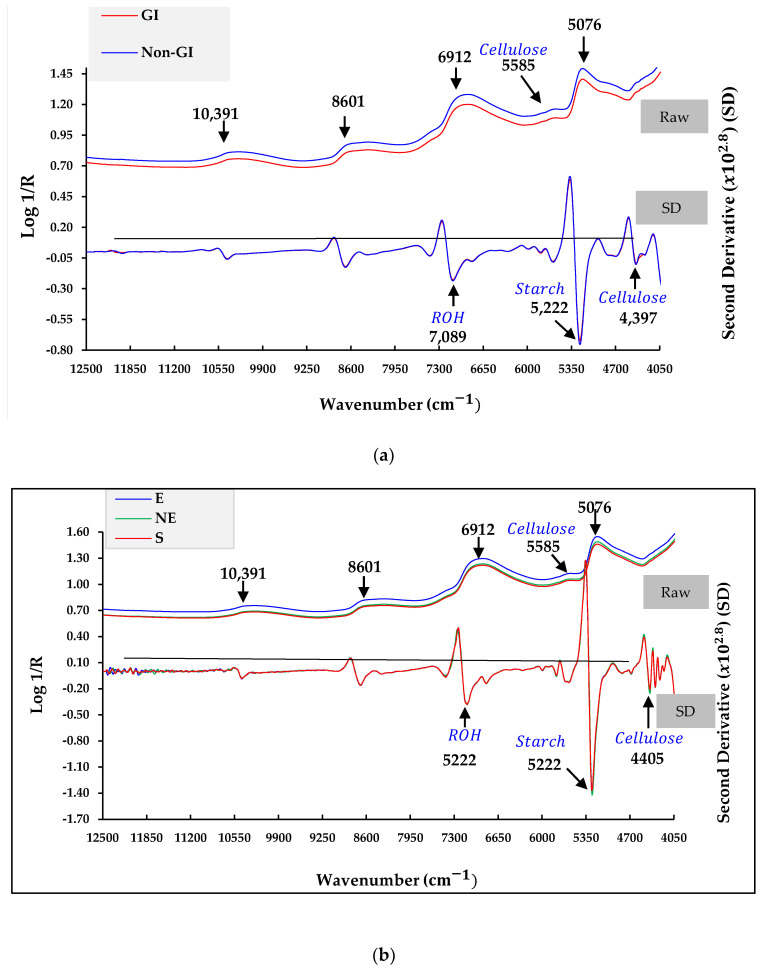
Average raw (Raw) and Savitzky-goley second derivative (SD) pre-processing spectral data of intact peel (with thorns) (I-form) (**a**) Average raw (Raw), Baseline offset (Baseline), and SD pre-processing spectral data of a combination of intact peel (with thorns), cut-thorns peels, and stems (ICS-form) (**b**).

**Table 1 foods-12-03844-t001:** Location of durian samples.

Harvesting Time	Harvesting Area		Latitude (°)	Longitude (°)	Number of Samples	Geographical Origin	
	Province	Abbreviations				R *	GI *
May 2019	Chanthaburi	Chan	13.566599	100.581634	15	East	Non-GI
May 2019	Prajinburi	PraJ	10.986056	102.488185	10	East	Non-GI
May 2019	Rayong	Ra	12.999152	101.216687	15	East	Non-GI
May 2019	Trat	T	13.562512	100.260605	15	East	Non-GI
June 2019 and June 2020	Prachuap khiri khan	PraCh	10.986058	102.488186	25	South	GI
August 2019	Chumphon	Chum	12.116060	100.346539	5	South	Non-GI
July 2020	Surat thani	Su	10.986058	102.488186	10	South	Non-GI
June 2019 and June 2020	Sri sa kat	Sri	15.024017	103.438389	25	North East	Non-GI

* R: region, GI: geographical identification.

**Table 2 foods-12-03844-t002:** The absorption bands of durian samples in three forms in common: intact peel (with thorns) (I-form), cut-thorns peel (C-form), and stem (S-form).

Wavenumber (cm^−1^)	Wavelength (nm)		Bond Vibration	Possible Compound Reference	Ref.
	Sample	Referred from Ref.			
5122	1952	1940	O–H str.*+ O–H def.*	H_2_O	[36]
5631	1775	1780	C–H str.* first overtone	Cellulose	[36]
6827	1465	1450	O–H str.* first overtone	Starch, H_2_O	
8508	1175	1190	of heavy water, formally called deuterium oxide (2H_2_O or H_2_O)	H_2_O	[39]
10,314	970	970	O–H str.* first overtone	H_2_O	[36]

* str.: stretching, def.: deformation, ref.: reference.

**Table 3 foods-12-03844-t003:** Percent of accuracy (% accuracy) of effective model of geographical origin classification of durian compared using 24 algorithms in classification learner app of the effective models.

Geographical Origin	Form	Number of Spectra	Pre-Processing, Selected Wavenumbers	% Accuracy			Classification Learner Algorithm
				Data set			
		CV */Train */Test *		CV *	Train *	Test *	
GI *	I *	419/419/140	SNV	90.2	89.3	89.3	Neural Network
			SNV+GA	85.7	88.3	92.1	[Narrow Neural Network]
	SMOTE-I *	684/684/228	SNV	95.6	95.0	94.7	Neural Network
			SNV+GA	94.3	94.6	93.6	[Narrow Neural Network]
	C *	414/414/139	SNV	85.5	82.9	91.4	Neural Network
			SNV+GA	86.2	83.8	89.9	[Wide Neural Network]
	SMOTE-C *	674/674/226	Baseline	88.9	89.5	76.5	Neural Network
			Baseline+GA	87.2	85.6	79.6	Bilayered Neural Network
	S *	193/193/64	SNV	88.1	91.2	84.4	Discriminant Analysis
			SNV+GA	87.6	83.4	81.2	[Linear Discriminant]
	SMOTE-S *	314/314/104	MSC	93.3	93.3	95.2	Neural Network
			MSC+GA	91.7	94.9	90.4	[Medium Neural Network]
	IC *	883/883/279	SNV	86.0	85.8	90.7	Neural Network
			SNV+GA	86.8	84.3	91.0	[Medium Neural Network]
	SMOTE-IC *	1358/1358/454	MeanNor	94.6	94.3	92.3	Neural Network
			MeanNor+GA	91.5	91.0	88.1	[Bilayered Neural Network]
	IS *	611/611/205	SNV	88.2	90.3	91.2	SVM *
			SNV+GA	89.4	87.4	91.2	[Quadratic Discriminant]
	SMOTE-IS *	996/996/334	MSC	88.2	90.3	91.2	Neural Network
			MSC+GA	94.0	91.4	89.5	[Trilayered Neural Network]
	CS *	608/608/202	Raw	78.2	88.0	84.7	SVM *
			Raw+GA	85.4	85.9	83.7	[Cubic SVM *]
	SMOTE-CS *	990/990/328	Baseline	93.9	92.2	82.9	Neural Network
			Baseline+GA	91.8	90.8	77.7	[Trilayered Neural Network]
	ICS *	1026/1026/343	MSC	87.5	86.5	92.1	SVM *
			MSC+GA	82.8	83.2	89.8	[Cubic SVM *]
	SMOTE-ICS *	1672/1672/558	MSC	93.9	93.2	90.3	SVM
			MSC+GA	92.2	91.5	87.6	[Cubic SVM *]
R*	I *	419/419/140	Raw	64.0	64.2	74.3	Neural Network
			Raw+GA*	61.3	59.9	71.4	[Wide Neural Network]
	SMOTE-I *	549/549/183	SNV*	77.4	80.5	80.9	SVM*
			SNV*+GA*	80.1	78.9	78.1	[Quadratic Discriminant]
	C *	419/419/140	MeanNor*	76.6	76.1	91.4	Neural Network
			MeanNor*+GA*	71.5	72.2	85.6	[Bilayered Neural Network]
	SMOTE-C *	534/534/180	MeanNor*	85.6	84.5	92.8	Neural Network
			MeanNor+GA*	80.5	82.4	83.9	[Medium Neural Network]
	S *	193/193/64	Raw	63.2	61.1	87.5	Ensemble classifiers
			Raw+GA*	61.1	60.6	79.7	[Subspace Discriminant]
	SMOTE-S *	246/246/84	SD*	56.1	57.3	88.1	SVM*
			SD*+GA*	60.2	59.3	81.0	[Quadratic Discriminant]
	IC *	833/833/279	MeanNor*	76.7	78.2	81.0	Neural Network
			MeanNor*+GA*	76.5	77.9	78.5	[Narrow Neural Network]
	SMOTE-IC *	1083/1083/363	MeanNor*	87.4	86.6	79.3	Neural Network
			MeanNor+GA*	80.7	80.3	77.7	[Narrow Neural Network]
	IS *	611/611/205	SNV*	73.3	72.8	75.6	Neural Network
			SNV+GA*	65.0	64.6	71.7	[Narrow Neural Network]
	SMOTE-IS *	795/795/267	SNV*	79.7	80.6	84.3	SVM*
			SNV*+GA*	77.9	77.4	80.9	[Cubic SVM*]
	CS *	608/608/202	MeanNor*	81.1	78.9	88.6	Neural Network
			MeanNor*+GA*	75.3	78.5	86.1	[Narrow Neural Network]
	SMOTE-CS *	786/786/261	MeanNor*	86.6	85.5	90.4	Neural Network
			MeanNor*+GA*	87.4	86.0	82.4	[Narrow Neural Network]
	ICS *	1026/1026/343	Baseline*	68.1	67.9	71.1	Neural Network
			Baseline*+GA*	65.2	63.7	67.9	[Narrow Neural Network]
	SMOTE-ICS *	1332/1332/444	SNV*	88.0	86.5	80.0	Ensemble classifiers
			SNV*+GA*	64.5	65.3	57.2	[Subspace Discriminant]

* GI: geographical identification, 3R: Region, CV: Cross validation set (using five k-folds), Train: Training set, Test: Testing set, GA: genetic algorithm, SMOTE: Synthetic Minority Over-sampling Technique, I: intact peel (with thorns) (I-form), C: cut-thorns peel (C-form), S: stem (S-form), I+C: IC-form, I+S: IS-form, C+S: CS-form, I+C+S: ICS-form, Raw: no pre-processing, SNV: Standard normal variate, Baseline: Baseline offset, MSC: Multiplicative scattering correction, SD: Savitzky-Goley second derivative, MeanNor: Mean-normalization.

**Table 4 foods-12-03844-t004:** The confusion matrix and classification performance of the best effective models of geographical origin classification of durian.

Geographical Origin	Form	Pre-Processing	Classification Learner Algorithm	Number of Spectra	Data Set	%ACC	Prediction Class				TP *	TN *	FN *	FP *	%Precision	%Sensitivity	%Specifity	F1 Score
							Class	GI *	Non-GI *									
**GI ***	SMOTE-I *	SNV *	Neural Network	684	cv *	95.60	GI*	335	7		335	319	7	23	93.58	97.95	93.27	95.71
			[Narrow Neural Network]				Non*	23	319		319	335	23	7	97.85	93.27	97.95	95.51
				684	train *	95.00	GI *	333	9		333	317	9	25	93.02	97.37	92.69	95.14
							Non *	25	317		317	333	25	9	97.24	92.69	97.37	94.91
				228	test *	94.70	GI *	107	7		107	109	7	5	95.54	93.86	95.61	94.69
							Non	5	109		109	107	5	7	93.97	95.61	93.86	94.78
							**Class**	**E ***	**NE ***	**S ***								
R *	SMOTE-ICS *	SNV *	Ensemble classifiers	1332	cv *	88.00	E*	380	25	39	380	792	64	25	93.83	85.59	96.94	89.52
			[Subspace Discriminant]				NE*	3	429	12	429	743	15	84	83.63	96.62	89.84	89.66
							S*	22	59	363	363	809	81	51	87.68	81.76	94.07	84.62
				1332	train *	86.50	E *	370	33	41	370	782	74	32	92.04	83.33	96.07	87.47
							NE *	1	425	18	425	727	19	89	82.68	95.72	89.09	88.73
							S *	31	56	357	357	795	87	59	85.82	80.41	93.09	83.02
				444	test *	80.00	E *	138	4	6	138	217	10	20	87.34	93.24	91.56	90.20
							NE *	16	95	37	95	260	53	26	78.51	64.19	90.91	70.63
							S *	4	22	122	122	233	26	43	73.94	82.43	84.42	77.96

* GI: Geographical identification, R: Region, I: Intact peel (with thorns), ICS: intact peel (with thorns) (I-form)+cut-thorns peel (C-form)+ stem (S-form), SNV: Standard normal variate, CV: Cross validation set (using five k-folds), Train: Training set, Test: Testing set, %ACC: %Accuracy SMOTE: Synthetic Minority Over-sampling Technique, GI: Geographical identification class, Non-GI: Not geographical identification class, TP: true positive, TN: true negative, FP: false positive, FN: false negative.

**Table 5 foods-12-03844-t005:** Modelling parameters.

Method	Parameter
SMOTE	Number of nearest neighbors = 5
Neural Network classifiers	Preset: Narrow Neural Network
	Number of fully connected layers: 1
	First layer size: 10
	Activation: ReLU *
	Iteration limit: 1000
	Regularization strength (Lamnda): 0
	Standardize data: Yes
Ensemble classifiers	Preset: Subspace Discriminant
	Ensemble method: Subspace
	Learner type: Discriminant
	Number of learners: 30
	Subspace dimension: 577

* ReLU: activation operation performs a nonlinear threshold operation, where any input value less than zero is set to zero.

## Data Availability

Data are contained within the article.

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
