# Peer review of "A Geographical Origin Classification of Durian (cv. Monthong) Using Near-Infrared Diffuse Reflectance Spectroscopy"

_foods, 2023, doi:10.3390/foods12203844_

Round 1
Reviewer 1 Report
The manuscript is written with clear understanding of the project addressed, but there are some problems. It is recommended to review after major revisions to the manuscript. My specific comments are as follows:
1、In line 92-93, the discrimination of durian geographical origin is limited to the detection of a few active components with analytical methods. The analytical methods must be specific.
2、There is a problem with the labeling of references.
3、Figure 3, the datasets were randomly divided into training and test sets at a 33:1 ratio, what is the basis for the division? According to the current situation, 7:3 and 6:4 are the general division ratios.
4、In abstract, line 23, 64.90% Accuracy by testing set were obtained. The Accuracy is too low.
Author Response
|
Response to Reviewer 1 Comments |
||
|
1. Summary |
|
|
|
Reviewer 1 The manuscript is written with a clear understanding of the project addressed, but there are some problems. It is recommended to review after major revisions to the manuscript. Thanks so much for your kindness in allowing us the opportunity to revise our manuscript. According to the reviewer’s comments, we can make significant improvements to our work. We are truly grateful. If there's an answer I can't provide well, please kindly suggest us. |
||
|
2. Questions for General Evaluation |
Reviewer’s Evaluation |
Response and Revisions |
|
Does the introduction provide sufficient background and include all relevant references? |
Yes/Can be improved/Must be improved/Not applicable |
|
|
Are all the cited references relevant to the research? |
Yes/Can be improved/Must be improved/Not applicable |
|
|
Is the research design appropriate? |
Yes/Can be improved/Must be improved/Not applicable |
|
|
Are the methods adequately described? |
Yes/Can be improved/Must be improved/Not applicable |
|
|
Are the results clearly presented? |
Yes/Can be improved/Must be improved/Not applicable |
|
|
Are the conclusions supported by the results? |
Yes/Can be improved/Must be improved/Not applicable |
|
|
3. Point-by-point response to Comments and Suggestions for Authors |
||
|
Comments 1: In line 92-93, the discrimination of durian geographical origin is limited to the detection of a few active components with analytical methods. The analytical methods must be specific. |
||
|
Response 1: analytical methods are the stable isotope and element compositions analysis. I added in a new attachment. |
||
|
Comments 2: There is a problem with the labeling of references. |
||
|
Response 2: I revised as new attachment. |
||
|
Comments 3: Figure 3, the datasets were randomly divided into training and test sets at a 33:1 ratio, what is the basis for the division? According to the current situation, 7:3 and 6:4 are the general division ratios. |
||
|
Response 3: we used all spectra and divide ratio (75:25). We used Kennard-Stone method, it outperforms the random sampling in the selection of calibration samples. |
||
|
Comments 4: In abstract, line 23, 64.90% Accuracy by testing set were obtained. The Accuracy is too low. |
||
|
Response 4: The sentence is revised in ABSTRACT by including the phrase - indicating the classification model of East (E-region), Northeast (NE-region) and South (S-region) regions could be applied for rough screening. – and in conclusion with reference [51]. Williams, P.; Manley, M.; Antoniszyn, J., Near infrared technology: getting the best out of light. African Sun Media: 2019. |
||
|
4. Response to Comments on the Quality of English Language |
||
|
Point 1: I am not qualified to assess the quality of English in this paper |
||
|
Response 1: |
||
|
5. Additional clarifications |
||
|
‘- |
||

Reviewer 2 Report
I have reviewed the manuscript entitled “A geographical origin classification of durian (cv. Monthong) using near-infrared diffuse reflectance spectroscopy”. This manuscript lacks methodology from both the experimental design and the analytical point of view. The novelty and significance of this work should be made more explicit, especially in the abstract and conclusion sections. There is no sufficient discussion of the results obtained.
Detailed remarks:
Avoid including keywords already present in the title.
The Introduction is too long and should be revised to focus on the topic under study.
Location of Durian samples: Why these regions were chosen should be explained in detail. The importance of the selected regions in the country and the world in terms of production should be explained. Describe the sampling process thoroughly. Variations in agroecological conditions, annual weather patterns, cultural practices, and the number of samples analyzed can affect the results. Clarify how homogeneity and traceability were maintained, and provide details on the number of replicates taken and analyzed.
The Results and Discussion section contains many tables, but the results lack in-depth discussion. Explain the trends observed in your results within this section. Provide more interpretation.
In conclusion, emphasize your work's novelty, utility, and implications and the significance of your results.
The language needs improvement throughout the manuscript to reflect a more scientific perspective.
The manuscript is not prepared in accordance with the guidelines of the journal. Especially the references section needs to be revised in terms of the journal's guidelines.
The language needs improvement throughout the manuscript to reflect a more scientific perspective.
Author Response
|
Response to Reviewer 2 Comments |
||
|
1. Summary |
|
|
|
Reviewer 2 I have reviewed the manuscript entitled “A geographical origin classification of durian (cv. Monthong) using near-infrared diffuse reflectance spectroscopy”. This manuscript lacks methodology from both the experimental design and the analytical point of view. The novelty and significance of this work should be made more explicit, especially in the abstract and conclusion sections. There is no sufficient discussion of the results obtained. Thanks so much for your kindness in allowing us the opportunity to revise our manuscript. According to the reviewer’s comments, we can make significant improvements to our work. We are truly grateful. If there's an answer I can't provide well, please kindly suggest us. |
||
|
2. Questions for General Evaluation |
Reviewer’s Evaluation |
Response and Revisions |
|
Does the introduction provide sufficient background and include all relevant references? |
Yes/Can be improved/Must be improved/Not applicable |
|
|
Are all the cited references relevant to the research? |
Yes/Can be improved/Must be improved/Not applicable |
|
|
Is the research design appropriate? |
Yes/Can be improved/Must be improved/Not applicable |
|
|
Are the methods adequately described? |
Yes/Can be improved/Must be improved/Not applicable |
|
|
Are the results clearly presented? |
Yes/Can be improved/Must be improved/Not applicable |
|
|
Are the conclusions supported by the results? |
Yes/Can be improved/Must be improved/Not applicable |
|
|
3. Point-by-point response to Comments and Suggestions for Authors |
||
|
Comments 1: Avoid including keywords already present in the title. |
||
|
Response 1: Title: A geographical origin classification of durian (cv. Monthong) using near-infrared diffuse reflectance spectroscopy Changing: Geographical identification (GI); Regions classification; Near-infrared spectroscopy; Durian; Classification; Synthetic minority over-sampling technique [change in red text, attached] |
||
|
Comments 2: The Introduction is too long and should be revised to focus on the topic under study. |
||
|
Response 2: agree. I revised as in revised manuscript. |
||
|
Comments 3: Location of Durian samples: Why these regions were chosen should be explained in detail. The importance of the selected regions in the country and the world in terms of production should be explained. Describe the sampling process thoroughly. Variations in agroecological conditions, annual weather patterns, cultural practices, and the number of samples analyzed can affect the results. Clarify how homogeneity and traceability were maintained and provide details on the number of replicates taken and analyzed. |
||
|
Response 3: Figure 1 shows different locations along level of latitude and longtitude. East; Prajinburi, Rayong, Chanthaburi and Trat. North east; Si sa ket and South; Surat thani, Chumphon and Prachuap khiri khan, which has a direct effect on geographic characteristics. The original durian shows differences in taste, texture, and flavor. Durian that comes from different provinces influences consumers and price. The harvesting period of provinces or regions might overlap. Therefore, the sampling process was selected to follow durian yield in each province (according to an Office of Agricultural Economics report) This is why these regions were chosen for this research. The variations in agroecological conditions, annual weather patterns, cultural practices, and the number of samples affected of result which is the main aim of this study to using NIRs technique to classify different groups of samples whether it is in different geographical identification or regions. For the sampling process, we manage the variation of sampling by making the following. 1) Collected 3 orchards per province. 2) The durian tree age of 5-10 years which is the production year. 3) 2 years for collecting samples (2019-2020) indicated the replication of sampling to include variation effect of agroecological conditions and annual weather patterns. 4) all durians were tag 120 DAA (day after anthesis) which is the commercial harvesting date. The tagging sampling was 3 durians and selected 1. 5) Durian sample was tagged in middle level of canopy. 6) Transportation was controlled within 3 days and the fruit samples were controlled for uniformed temperature before scanning to avoid spectral error of experiment. 7) Repeating the scan three times at the same point and using all the data in the data analysis. Homogeneity and traceability if mean no variation in durian sample of each location it was clarified and confirmed by this sampling management. |
||
|
Comments 4: The Results and Discussion section contains many tables, but the results lack in-depth discussion. Explain the trends observed in your results within this section. Provide more interpretation. In conclusion, emphasize your work's novelty, utility, and implications and the significance of your results. |
||
|
Response 4: The Results and Discussion section contains many tables, but the results lack in-depth discussion. Explain the trends observed in your results within this section. Provide more interpretation. I have included the result and discussion in those tables [Add in blue text, in revised manuscript] In conclusion, emphasize your work's novelty, utility, and implications and the significance of your results. I revised as last paragraph to conclude the novelty, its utility and implications, and the significance of the results. [Add in blue text, in revised manuscript] |
||
|
Comments 5: The language needs improvement throughout the manuscript to reflect a more scientific perspective. The manuscript is not prepared in accordance with the guidelines of the journal. Especially the references section needs to be revised in terms of the journal's guidelines. |
||
|
Response 5: . I will submit our revised manuscript to MDPI English Editing after we get revision for the second round or if our manuscript is accepted. I have revised manuscript to be in accordance with the guidelines of the journal, especially the references section. |
||
|
4. Response to Comments on the Quality of English Language |
||
|
Point 1: The language needs improvement throughout the manuscript to reflect a more scientific perspective. |
||
|
5. Additional clarifications |
||
|
- |
||

Reviewer 3 Report
The manuscript addresses the issue of geological provenance. It describes in detail and with precision the experimental conditions, the recording of spectra and the chemometric procedures used.
A detailed overview of the different classification models is given.
The manuscript is also formally correct: clear data, informative figures, explanations of abbreviations are never missing.

Author Response
|
Response to Reviewer 3 Comments |
||
|
1. Summary |
|
|
|
Reviewer 3 The manuscript addresses the issue of geological provenance. It describes in detail and with precision the experimental conditions, the recording of spectra and the chemometric procedures used. A detailed overview of the different classification models is given. The manuscript is also formally correct: clear data, informative figures, explanations of abbreviations are never missing. Thanks so much for your kindness in allowing us the opportunity to revise our manuscript. According to the reviewer’s comments, we can make significant improvements to our work. We are truly grateful. If there's an answer I can't provide well, you suggest it again. |
||
|
2. Questions for General Evaluation |
Reviewer’s Evaluation |
Response and Revisions |
|
Does the introduction provide sufficient background and include all relevant references? |
Yes/Can be improved/Must be improved/Not applicable |
|
|
Are all the cited references relevant to the research? |
Yes/Can be improved/Must be improved/Not applicable |
|
|
Is the research design appropriate? |
Yes/Can be improved/Must be improved/Not applicable |
|
|
Are the methods adequately described? |
Yes/Can be improved/Must be improved/Not applicable |
|
|
Are the results clearly presented? |
Yes/Can be improved/Must be improved/Not applicable |
|
|
Are the conclusions supported by the results? |
Yes/Can be improved/Must be improved/Not applicable |
|
|
3. Point-by-point response to Comments and Suggestions for Authors |
||
|
Comments 1: The form of the bibliographical references in the text is incorrect. References should be inserted in the form [x], not subscript. I recommend using reference management programs (Zotero, Mendeley, etc.). |
||
|
Response 1: I revised as new attachment. |
||
|
Comments 2: Figure 6 is too cluttered for me. I don't understand the caption in Figure 6b: what does "Regions regions" mean. Since Table 3 contains the information presented in Figure 6, I feel Figure 6 is redundant. |
||
|
Response 2: I agreed and Edited “Regions regions” to “Regions (R). And deleted Figure 6 because it is the same as Table 3 |
||
|
4. Response to Comments on the Quality of English Language |
||
|
Point 1: I am not qualified to assess the quality of English in this paper |
||
|
Response 1: - |
||
|
5. Additional clarifications |
||
|
‘- |
||

Round 2
Reviewer 1 Report
Well done!
Reviewer 3 Report
The authors have corrected the problems raised.
I accept the corrected version and recommend the manuscript for publication